# Gut Microbiota-Derived l-Histidine/Imidazole Propionate Axis Fights against the Radiation-Induced Cardiopulmonary Injury

**DOI:** 10.3390/ijms222111436

**Published:** 2021-10-23

**Authors:** Zhiyuan Chen, Bin Wang, Jiali Dong, Yuan Li, Shuqin Zhang, Xiaozhou Zeng, Huiwen Xiao, Saijun Fan, Ming Cui

**Affiliations:** 1Tianjin Key Laboratory of Radiation Medicine and Molecular Nuclear Medicine, Institute of Radiation Medicine, Chinese Academy of Medical Sciences and Peking Union Medical College, Tianjin 300192, China; Chen_ZYuan@163.com (Z.C.); wangbinpumc@126.com (B.W.); dongjiali66@126.com (J.D.); liyuan@irm-cams.ac.cn (Y.L.); zhangshuqin@irm-cams.ac.cn (S.Z.); zengxztj@student.pumc.edu.cn (X.Z.); fansaijun@irm-cams.ac.cn (S.F.); 2Department of Microbiology, College of Life Sciences, Nankai University, Tianjin 300071, China

**Keywords:** radiation-induced cardiopulmonary injury, gut microbiota metabolites, imidazole propionate, pyroptosis, l-histidine

## Abstract

Radiation-induced cardiopulmonary injuries are the most common and intractable side effects that are entwined with radiotherapy for thorax cancers. However, the therapeutic options for such complications have yielded disappointing results in clinical applications. Here, we reported that gut microbiota-derived l-Histidine and its secondary metabolite imidazole propionate (ImP) fought against radiation-induced cardiopulmonary injury in an entiric flora-dependent manner in mouse models. Local chest irradiation decreased the level of l-Histidine in fecal pellets, which was increased following fecal microbiota transplantation. l-Histidine replenishment via an oral route retarded the pathological process of lung and heart tissues and improved lung respiratory and heart systolic function following radiation exposure. l-Histidine preserved the gut bacterial taxonomic proportions shifted by total chest irradiation but failed to perform radioprotection in gut microbiota-deleted mice. ImP, the downstream metabolite of l-Histidine, accumulated in peripheral blood and lung tissues following l-Histidine replenishment and protected against radiation-induced lung and heart toxicity. Orally gavaged ImP could not enter into the circulatory system in mice through an antibiotic cocktail treatment. Importantly, ImP inhibited pyroptosis to nudge lung cell proliferation after radiation challenge. Together, our findings pave a novel method of protection against cardiopulmonary complications intertwined with radiotherapy in pre-clinical settings and underpin the idea that gut microbiota-produced l-Histidine and ImP are promising radioprotective agents.

## 1. Introduction

Cancer ranks as the top public enemy and is the illness with the highest incidence and largest patient number in the world. Because of technological advancements, the number of cancer survivors is gradually reaching 33 million [1]. In developed countries, radical medical resources result in 67% of cancer patients surviving for more than 5 years, with about 25% of patients surviving for more than 15 years. However, cancer survivors often suffer from adverse side effects that are ineluctably intertwined with treatments such as radiotherapy, chemotherapy, or immunotherapy [2]. As a first-line therapeutic option for multiple kinds of cancer, about 50% of patients with solid tumors require radiotherapy for curative treatment [3]. Although advances in delivery technology have transformed the remedy into a more precise treatment, radiotherapy is still entwined with short-term side effects such as mucositis and long-term side effects such as tissue fibrosis [4,5]. Radiotherapy for thoracic tumors often burdens the lungs and heart and leads to complications, such as pneumonia, cardiac pericarditis, cardiomyopathy, and pulmonary and myocardial fibrosis, which can occur months to years after radiotherapy [6]. In light of these orchestrated intercorrelations, the lungs and heart are collectively considered as the cardiopulmonary system. For instance, local lung irradiation causes pathologic changes in the heart and vice versa [7]; however, the underlying mechanisms are diversiform and sophisticated [8,9]. Therefore, the loss of cardiopulmonary function is further aggravated when both the lungs and heart are exposed to irradiation [10,11].

The gut microbiome plays important roles in human health [12]. Recently, burgeoning experimental and epidemiological evidence has highlighted the substantive intercorrelations between intestinal flora and the lungs, which has been termed as the “gut–lung axis” [13]. Gut microbiota-derived components, such as short-chain fatty acids (SCFA), penetrate into the circulatory system and serve as key messengers to connect the gut and lungs [10,14,15]. The crosstalk between host and gut microorganisms as well as the interactions within the gut microorganisms themselves induce short- and long-term effects. The gut–lung axis can shape the immune response in lung tissues and can interfere with the process of respiratory diseases [16]. Gut dysbiosis not only precipitates digestive tract diseases such as inflammatory bowel disease but also renders chronic obstructive pulmonary disease and asthma14. In addition, mounting evidence has proven that the gut microbiome is involved in cardiovascular disease, and changes in gut microbiome have been reported to cause atherosclerosis [17]. In the “gut–heart axis”, the metabolites produced by the gut microbiota also serve as essential linkers. For example, SCFAs are recognized as safeguards for cardiovascular diseases, while trimethylamine-N-oxide is considered to be an accessary to elicit cardiovascular diseases [15,18].

In this study, we aimed to screen bioactive compounds from gut microbiota-derived metabolites to battle against radiation-induced cardiopneumatic injuries. Our observations demonstrated that the l-Histidine/ImP axis lulled the pathological processes of the lung and heart tissues and ameliorated lung respiratory and heart systolic functions following radiation stimuli in a gut microbiota-dependent fashion. Mechanistically, ImP blocked pyroptosis to propel the proliferation of pneumonocytes. Together, our findings provide novel insights into the function and the underlying protective mechanism of microbiota metabolites in the context of radiation-induced cardiopneumatic toxicity in preclinical settings.

## 2. Results

### 2.1. Gut Microbiota-Derived l-Histidine Improves Radiation Toxicity in Lung and Heart

In light of our previous study, total chest irradiation (TCI) decreased the level of l-Histidine in fecal pellets, which was reversed following fecal microbiota transplantation (FMT, Appendix A). Given the remedial effects of the gut microbiome on radiation pneumonic toxicity, we treated the local chest-irradiated mice with l-Histidine via the oral route for 10 days, and the sham radiation and TCI-only group were given equal doses of placebo. As shown in Figure 1A, l-Histidine replenishment prevented weight loss, implying that l-Histidine might have had the potential to improve the radiotherapy prognosis for thorax cancers. Then, we assessed the levels of l-Histidine in feces, serum, and lung and observed that l-Histidine accumulated in the three parts following oral gavage (Figure 1B and Appendix A). Next, Sirius Red and Masson staining revealed the structural damage and collagen accumulation of the alveoli in irradiated mice, while l-Histidine attenuated the pathological process (Figure 1C,D). l-Histidine also erased the elevation of the lung coefficient following TCI (Figure 1E), suggesting that l-Histidine might improve lung function after radiotherapy. Indeed, l-Histidine treatment increased the respiratory quotient (RQ) value, decreased VO_2_ intake emissions, and alleviated pneumonia, as judged by the reduction of radiation-elevated pro-fibrotic and pro-inflammatory factors (Figure 1F–H and Appendix A). In line with the above observations, l-Histidine also hindered the structural damage and collagen accumulation in heart tissues (Figure 1I,J), recovered systolic and diastolic pressure (Figure 1K and Appendix A), and lessened the inflammatory status of the heart tissues in mice with local chest irradiation (Figure 1L and Appendix A). Together, our observations demonstrate that l-Histidine protects against radiation toxicity in both lung and heart tissues.

### 2.2. l-Histidine Shapes the Gut Microbiota Configuration after Chest Local Irradiation

Owing to the enrichment of l-Histidine in the fecal pellets, 16S rRNA sequencing was performed to analyze whether l-Histidine impacted gut microbiota with radiation stimuli. Local chest irradiation heightened the α-diversity of the intestinal bacteria, while l-Histidine erased the elevation (Figure 2A,B and Appendix A). Although both weighted and unweighted unifrac analyses showed no significant changes in the β-diversity (Figure 2D and Appendix A), and principal component analysis (PCA) and unweighted/weighted principal coordinate analysis (PCoA) validated the separations of the gut microbiota composition among the three cohorts (Figure 2C,E,F), indicating that l-Histidine indeed shifts the gut microbiota community in irradiated mice. In detail, the TCI-exposed mice showed a reduction in Akkermansia_muciniphila, Lactobacillus_reuteri, Clostridium_sp_Cuiture-41, lachnospiraceae_bacrerium_615, Ileibacterium_valens, and Helicobacter_bills at the species level, which were enriched following l-Histidine replenishment (Figure 2G). Together, all of the evidence seen here bolsters that l-Histidine treatment restructures the gut microbiota configuration in local chest-irradiated mice.

### 2.3. Gut Microbiota Contributes to l-Histidine-Mediated Radioprotection

Next, we employed antibiotic-treated, gut microbiota-deleted mice to explore the role of intestinal flora in l-Histidine in terms of how it improved radiation pneumonic toxicity. In the first model, both groups were administrated l-Histidine, and an antibiotic cocktail (ABX) was added to the drinking water for one of the two groups. As shown by the Sirius Red and Masson staining, gut microbiota deletion interfered with l-Histidine-protected induced structural damage and collagen accumulation in local chest-irradiated mice (Figure 3A). In addition, ABX treatment also up-regulated the expression of pro-fibrotic and pro-inflammatory cytokines in lung tissues (Figure 3B–D). In the second model ABX was added to the drinking water for the other two cohorts, and only one cohort was treated with l-Histidine. In parallel with the first model, l-Histidine replenishment failed to protect radiation-induced pneumonic injury when the gut microbiota was not present, as judged by the un-elevated body weight (Figure 3E) and RQ value (Figure 3F) as well as unchanged pro-fibrotic and pro-inflammatory factors (Figure 3G–I). All of the evidence manifests that l-Histidine protects against radiation-induced pneumonic injury depending on gut microbiota, at least partly.

### 2.4. l-Histidine Remolds the Gut Microbiota Metabolome Fluctuated by Local Chest Irradiation

To further address the effects of l-Histidine on the gut microbiota after TCI, we assessed the metabolome of enteric flora using UHPLC-MS/MS. The heatmap described the details of metabolome changes. For example, TCI increased the relative abundance of 4-(acetylamino)phenyl 3-chlorobenzoate and abscisic acid but decreased that of l-aspartic acid (Figure 4A). However, l-Histidine treatment elevated the frequency of prostaglandin f1α and reduced that of testosterone sulfate (Figure 4B). All of the results further validated that TCI shaped the gut microbiota patten, which was restructured by l-Histidine replenishment. Given that l-Histidine is catabolized into ImP or histamine to favor physiological processes [19], we further assessed the levels of ImP and histamine in serum and lung tissues by ELISA. The results showed that only ImP and not histamine accumulated in serum and lung tissue following l-Histidine gavaging (Figure 4C–F), suggesting that ImP might be the key downstream metabolite of l-Histidine to protect the lungs and heart against irradiation.

### 2.5. Imidazole Propionate Ameliorates Chest Local Irradiation-Induced Toxicity

Next, we treated local chest-irradiated mice with ImP for 10 days and obtained that ImP was accumulated in the lungs after oral gavaging (Figure 5B). Intriguingly, ImP administration impeded the weight loss of irradiated mice and prevented structural damage to the lung and collagen accumulation in alveoli (Figure 5A,C,D). Consistent with l-Histidine, ImP increased the RQ value and down-regulated the levels of pro-fibrotic and pro-inflammatory cytokines in the lung tissues and bronchoalveolar lavage fluid (BALF) following radiation challenge, indicating that ImP, a secondary metabolite of l-Histidine, is able to fight against radiation-induced pneumonic injury (Figure 5E–H and Appendix A). We also assessed radiation cardiac toxicity in this system. Sirius Red and Masson staining showed that ImP attenuated local chest radiation-induced structural damage and collagen accumulation in heart tissues (Figure 5I,J). In addition, oral gavaging of ImP restored systolic and diastolic pressure, reduced the heart inflammatory status of irradiated mice (Figure 5K,L and Appendix A). Together, our observations demonstrate that l-Histidine-derived ImP is a key element to protect against radiation-induced pneumonic and cardiac toxicity.

### 2.6. Gut Microbiota Impacts the Assimilation of Imidazole Propionate

In light of the above results, l-Histidine might be catabolized into ImP or absorbed directly to perform radioprotective effects. Here, we used the aforementioned ABX treatment models to further assess whether ImP alleviates radiation toxicity relying on gut microbiota. In the first model, drinking water with ABX added to it interfered with the radioprotection of ImP to lung injury, as judged by lower RQ value and higher pro-fibrotic and pro-inflammatory factors (Figure 6A–E and Appendix A), inferring that the gut microbiota might bolster the radioprotective effects of ImP. In the second model, ABX-treated, gut microbiota-deleted mice did not respond to ImP implementation. In addition, ImP replenishment did not change the body weight (Figure 6F) or the RQ value (Figure 6G). Sirius Red and Masson staining revealed the similar structural damage and collagen accumulation in the alveoli caused by the irradiated mice between the two groups (Figure 6H). The pro-fibrotic and pro-inflammatory cytokines levels (Figure 6I,J and Appendix A) were also assessed in the gut microbiota-deleted mice. Finally, we assessed the ImP levels in the lung and serum in the mice who had undergone ABX treatment. Notably, the addition of ImP could not reach to peripheral blood and lung tissues without enteric bacteria (Figure 6K,L). We further used vancomycin or streptomycin to delete Gram-positive or Gram-negative bacteria, respectively, and observed that the accumulation of ImP in the lung tissue and serum was impaired in mice from the vancomycin treatment group (Appendix A), suggesting that Gram-positive bacteria may play an important role in the absorption of imidazole propionate. All of these findings bolster that mice with an aberrant gut microbiome cannot absorb imidazole propionate into the blood and lung tissues from the digestive tract, suggesting that the gut commensal microbiota might contribute to the radioprotection of ImP for cardiopulmonary injury.

### 2.7. Imidazole Propionate Inhibits Pyrolysis of Irradiated Lung Cells

We further explored the underlying mechanism by which ImP protected radiation-induced lung toxicity. Clone formation assays showed that both l-Histidine and ImP facilitated the proliferation of human bronchial epithelial cells (BEAS-2B) in a dose-dependent manner after radiation exposure; however, ImP represented more obvious radioprotection (Figure 7A). In light of the results, 8 μL/mL of ImP was used as the optimal concentration in the subsequent experiments. Pyroptosis plays a key role in lung injury [20]. Thus, we assessed whether ImP could inhibit the pyrotosis of lung cells following irradiation. The IHC results showed that local chest irradiation elevated the level of Gasdermin D (GSDMD) in lung tissue, a classic marker of pyrotosis, which was erased by ImP replenishment (Figure 7B), and qRT-PCR assay revealed that ImP decreased irradiation-increased GSDMD, TNF-α, and NF-κB expression in BEAS-2B cells, which was further validated by Western blotting, implying that Imp is able to lull radiation-precipitated pyroptosis in lung cells (Figure 7C–F). Pyroptosis is driven by caspase family activation and the cutting of GSDMD, resulting in the release of IL-1β, IL-18, and other inflammatory factors. Accordingly, the addition of ImP inhibited the activation of caspase-1, caspase-4, and caspase-5 in irradiated BEAS-2B cells (Figure 7G) and decreased IL-1β and IL-18 in the relative culture medium (Figure 7H,I). Immunofluorescence assay further showed that the addition of ImP elevated F-actin level to maintain the integrity of the cytoskeleton and to protect against mormorphic damage caused by pyroptosis (Figure 7J). Together, our observations demonstrate that ImP inhibits the pyroptosis of irradiated lung cells.

## 3. Discussion

Although high-energy ionizing radiation can kill cancer cells successfully, at the same time, it harms to the surrounding healthy tissues unavoidably and leads to various adverse side effects [21]. Radiotherapy has been widely used in thoracic malignancies, including breast cancer, lung cancer, esophageal cancer, thymoma, and Hodgkin’s lymphoma, which are located close to the heart and lungs anatomically [22]. During radiotherapy for thorax cancers, the heart and lungs may be at risk of being exposed to the irradiation field, producing multiple dose-dependent complications [23]. Despite the significant advances in the delivery technology used for radiotherapy, up to 30% of patients receiving thoracic radiotherapy still suffer from radiation-induced lung injury (RILI) [24]. Give the cumulative dose to the heart in radiotherapy for lung cancer, breast cancer, and lymphoma [25], radiation-induced heart disease is also common and worthy of attention. In a large population-based study, patients with left breast cancer who received radiotherapy were at 25% more risk of experiencing a cardiac event [26]. Radiation-induced cardiopulmonary injuries escalate as major dose limitation obstacles and degrade the life quality of cancer survivors. Heretofore, safe and effective remedies for these complications are still lacking in clinical scenarios.

Recent efforts have identified the close relationship between the gut microbiome and the cardiopulmonary system. For instance, fecal microbiota transplantation (FMT) has been reported to protect against radiation pneumonia [27]. However, FMT represents limitations in clinical applications, including aesthetic concerns, costs of donor screening, and material preparation and administration. Thus, screening radioprotective agents from the gut microbiome to substitute for FMT and fight against radiation toxicity is a novel and promising therapeutic avenue. Recently, mounting evidence have proven that gut microbiota-derived metabolites play important roles in lung injury and rehabilitation. For example, acetate dampens the activation of NLRP3 inflammasomes in neonatal mice with bronchopulmonary dysplasia and protects lung injury [28]; the L-tyrosine pathway, a metabolite produced by enteric microbes, impacts the airway epithelium at the distal end to reduce allergic airway reactions [29]. Phycocyanin is a light-harvesting pigment protein isolated from cyanobacteria that regulates the radiation-induced disturbance of the lung and intestinal flora of mice and reduces radiation-induced lung inflammation and fibrosis [30]. In this study, gut microbiota-derived l-Histidine drew our attention because the relative level of l-Histidine in fecal pellets decreased after local chest irradiation but increased after FMT. l-Histidine is an essential amino acid. The 2005 report of the American Institute of Medicine on diet recommends the intake of 4 g of l-Histidine daily [31]. l-Histidine has a unique role in proton buffering, metal ion chelation, the removal of reactive oxygen and nitrogen substances, and in erythropoiesis and histaminergic systems. l-Histidine has been proven to treat rheumatoid arthritis and anemia in patients with chronic renal failure [32]. l-Histidine was employed as an important component of preparation for organ preservation and myocardial protection in cardiac surgery [33]. In addition, studies have reported that carnosine (beta-alanyl-1-histidine) represents radioprotective effects on wound healing in rats [34] and can reduce radiation-induced lung damage [35]. In the present study, we screened out a single l-Histidine from intestinal microorganism-produced metabolites and found that oral gavage of l-Histidine educated and tuned the gut bacteria taxonomic proportions of local chest-irradiated mice. Importantly, the replenishment of l-Histidine via the oral route not only resisted radiation-induced respiratory dysfunction but also attenuated radiation-induced cardiac toxicity depending on the gut microbiota, at least partly.

l-Histidine can be catabolized into two classic secondary metabolites, imidazole propionate and histamine. Therefore, it is interesting to unravel whether l-Histidine protects against irradiation directly or relies on the downstream metabolites. To answer the question, we performed a battery of experiments. First, we used ABX to clean gut microbes. The gut microbiota-deleted mice did not respond to l-Histidine treatment and exhibited similar radiation-induced cardiopulmonary injuries to the chest of the local-irradiated controls. This suggests that destructing the metabolic function of the gut flora lull the radioprotective effects of l-Histidine. Secondly, we assessed the levels of ImP and histamine in PB and lung tissues from experimental mice with l-Histidine replenishment. The results show that only ImP was enriched in the target tissues. In addition, the oral gavage of ImP ameliorated radiation-induced cardiopulmonary toxicity, indicating that ImP might be the key secondary metabolites of l-Histidine to fight against radiation injuries. Finally, the experimental mice were treated with vancomycin or streptomycin, respectively. Results showed that the oral gavage of ImP cannot be enriched in PB and lung tissues in vancomycin-treated, Gram-positive bacteria-deleted mice, suggesting that gut commensal Gram-positive bacteria might play key roles in the absorption of ImP. Together, all of the evidence suggests that the gut microbiota might catabolize l-Histidine into ImP and assist in the entry of ImP into the circulatory system to protect against radiation-induced cardiopulmonary toxicity.

Acute pneumonia occurs within a few weeks after radiation and is characterized by the release of pro-inflammatory factors and the accumulation of immune cells in lung tissues. Research has reported that ionizing radiation precipitates pyroptosis in multiple tissues and organs (such as intestinal epithelium, liver, and muscle) [36]. Pyroptosis is programmed cell death, which is characterized as the formation of cell membrane pores, cytoplasmic swelling, membrane rupture, and the release of cytoplasmic contents (such as IL-1β) into the extracellular environment, amplifying local or systemic inflammatory responses [37]. Gasdermin D (GSDMD) is a major protein family that is related to pyroptosis and is involved in many inflammatory diseases and cancers [38]. Cleaved GSDMD forms transmembrane pores to enable the release of IL-1 and drive cell lysis through pyroptosis [39,40]. Caspase-1 also cleaves the inactive precursors in the IL-1 family to generate mature cytokines such as IL-1β and IL-18 [41,42]. In the present study, irradiation exposure elevated the expression of GSDMD in vitro and in vivo. In BEAS-2B cells, the expression of caspase-1, caspase-4, and caspase-5 were up-regulated after radiation stimuli, and the levels of IL-1β and IL-18 in culture medium were also heightened. In addition, the fluorescence intensity of the cytoskeleton protein F-actin was weakened and discontinuous after irradiation. All of these observations bolster the idea that irradiation nudged pyroptosis in lung cells after radiation challenge. Notably, ImP treatment inhibited the radiation-induced pyroptosis of lung cells. Given that NF-κB is an essential transcription factor for GSDMD [43,44]. ImP significantly blunted the increase of NF-κB after irradiation. It has been reported that the pre-treatment of glioma cells with a pyrrole-imidazole (Py-Im) polyamide prior to exposure to ionizing radiation resulted in a delay in the resolution of phosphorylated γ-H2AX foci that are indicative of the delayed repair of double-stranded breaks [45]. However, ImP, the classic downstream metabolite of l-Histidine, activated different signaling pathways to perform radioprotective functions in radiation-induced lung injury. Thus, our findings indicate that ImP fight against radiation-induced cardiopulmonary injuries through inhibiting pyroptosis.

## 4. Materials and Methods

### 4.1. Animals

Six- to eight-week-old-male (around 20 g) C57BL/6J mice were purchased from the Beijing Huafukang Bioscience Co., Inc. (Beijing, China). Mice were housed in the Specific Pathogen Free (SPF) level animal facility at the Institute of Radiation Medicine (IRM), the Chinese Academy of Medical Sciences (CAMS). All animal procedures were reviewed and approved by the Institutional Animal Care and Use Committee at CAMS.

### 4.2. Cell Culture

BEAS-2B cells were obtained from the American Type Culture Collection (ATCC) and were certified to be mycoplasma-free. The passage numbers of those cell lines during the experimental period were no more than eight. The cells were cultured with 10% fetal bovine serum (Gibco, Grand Island, NY, USA), 100 U/mL penicillin, and 100 mg/mL streptomycin and were grown at 5% CO_2_ and 37 °C.

### 4.3. Irradiation Study

A Gammacell−40 137Cs irradiator (Atomic Energy of Canada Limited, Chalk River, ON, Canada) at a dose rate of 0.8 Gy per minute was used for all experiments. Mice treated with total chest irradiation (TCI) were exposed to 15 Gy γ-ray, and control mice were sham-irradiated. Mice were intragastrically administered l-Histidine (Aladdin, CAS: 71-00-1, Shanghai, China) and imidazole propionate (Aladdin, 165812, CAS: 1074-59-5, Shanghai, China) at a dose of around 200 μL per mouse. Mice were sacrificed, and tissue samples were collected after the 21-day course of the experiment. BEAS-2B cells were exposed to 4 Gy (for colony formation assays) or 6 Gy (for other cell experiments).

### 4.4. Fecal Microbiota Transplantation (FMT)

For FMT administration, donor stool was freshly prepared on the day of transplant and within 4 h of transplantation. Donor stool was weighed and diluted with 2 mL of saline per 0.1 g of stool. In brief, the stool was steeped in saline for about 15 min, shaken, and then centrifuged at 800× *g* rpm for 3 min. The supernatant was obtained for treatment. Mice in the FMT group were gavaged with the supernatant 200 μL for 7 days before irradiation and then for 10 days after irradiation. TCI groups were given equal doses of placebo. The schematic diagram of FMT procedure is shown in the Appendix A.

### 4.5. Antibiotic Cocktail (ABX) Administration

ABX consisted of ciprofloxacin (0.125 g/L), metronidazole (0.1 g/L), vancomycin (0.05 g/L), streptomycin (100 U/L), and penicillin (100 U/L). Mice were exposed to ABX in drinking water after TCI. The same water was used for the water-treated (TCI + l-Histidine/ImP) group. Mice in the ABX group were housed with ABX fluid in their drinking water for 7 days before irradiation and then for 21 days after irradiation. The schematic diagram of the ABX procedure is shown in the Appendix A.

### 4.6. Enzyme-Linked Immunosorbent Assay (ELISA)

Frozen lung samples were ground up and reconstituted in PBS to a final concentration of 0.1 g/300 μL, followed by centrifugation for 10 min at 8000× *g* and 4 °C. Serum was collected using ethylenediaminetetraacetic acid (EDTA) as an anticoagulant. Serum was centrifuged for 15 min at 1000× *g* and 4 °C. Protein level was measured from the clear supernatant using an ELISA kit (Solarbio, Beijing, China) according to the manufacturer’s protocols. The supernatant was obtained to assess the level of ImP using the ELISA kit (F30264-A, mlBio, Shanghai, China) according to the manufacturer’s protocols (Zcibio, Shanghai, China). Optical density was read at 450 nm (Rayto, Shenzhen, China). (TNF-α, ml002095; TGF-β1, ml057830; IL-1, ml037875; mlBio, Shanghai, China.) The l-Histidine (or propionate imidazole) level in the lung tissues and serum were assessed in l-Histidine (or propionate imidazole) replenishment mouse models (mice with normal gut microbiota) and ABX-treated mouse models (gut microbiota-deleted mice).

### 4.7. Immunohistochemistry (IHC)

First, the slides were baked at 65 °C, and then the xylene solution was dewaxed, and different gradients of alcohol were dehydrated. After the dewaxing was completed, 3% H_2_O_2_ was added to the slide specimens. The slide was put into a 0.01 M citrate buffer (pH 6.0) pot and was heated at 95 °C for antigen retrieval to be performed. After being blocked with normal goat serum, the prepared primary antibody should continue to be dropped and incubated overnight in a refrigerator at 4 °C. On the second day, the biotin-labeled goat anti-rabbit secondary antibody working solution was incubated at room temperature for 10 min; the HRP-labeled horseradish peroxidase-labeled working solution was incubated at 37 °C for 10 min; the slices were dripped with newly configured DAB chromogenic solution; hematoxylin was counter-stained for 2 min; finally, the slides were dehydrated and mounted with neutral gum.

### 4.8. Colony Formation Assays

For colony formation analysis, 48 h after irradiation, 500 viable irradiated (transfected) cells were placed in 6-well plates and were maintained in complete medium for 2 weeks. Then, colonies were fixed with methanol and were stained with methylene blue.

### 4.9. Bronchoalveolar Lavage Fluid (BALF)

Mice were sacrificed, and one side of the bronchus was ligated, and the airway on the other side was lavaged three times with 800 μL ice-cold PBS. A total of 80% of the input volume was recovered, which was defined as BALF. BALF was centrifuged at 500× *g* for 10 min at 4 °C.

### 4.10. Respiratory Metabolism

We used the Panlab Oxylet system (Panlab, OXYLET, Spain) to monitor the 24-h RQ (O_2_/CO_2_), oxygen intake VO_2_ (mL/min/kg^0.75), and carbon dioxide emissions (VCO_2_ (mL/min/kg^0.75) of the mice in each group. Before monitoring, the system needed to be calibrated: low O_2_ (20%), high O_2_ (50%), low CO_2_ (0%), high CO_2_ (1.5%). Then, each mouse was weighed, and the average value was calculated. The mice were then placed in a metabolic cage. At this stage, the software should be opened in order to continuously monitor mice for 24 h, and the software should be used to calculate the relevant data.

VO_2_ and VCO_2_ weighted complete equations:(1)$$VO_{2}=\frac{\left(F\times \frac{{\left[O_{2}\right]}_{e}}{100}\right)-\frac{\frac{{\left[O_{2}\right]}_{s}}{100}\times F\times \left(1-\frac{{\left[O_{2}\right]}_{e}}{100}-\frac{{\left[CO_{2}\right]}_{e}}{100}\right)}{\left(1-\frac{{\left[O_{2}\right]}_{s}}{100}-\frac{{\left[CO_{2}\right]}_{s}}{100}\right)}}{W^{k}}$$
(2)$$VCO_{2}=\frac{\frac{F\times \left(1-\frac{{\left[O_{2}\right]}_{e}}{100}-\frac{{\left[CO_{2}\right]}_{e}}{100}\right)}{\left(1-\frac{{\left[O_{2}\right]}_{s}}{100}-\frac{{\left[CO_{2}\right]}_{s}}{100}\right)}\times \frac{{\left[CO_{2}\right]}_{s}}{100}-\frac{{\left[CO_{2}\right]}_{e}}{100}\times F}{W^{k}}$$
(3)$$\mathrm{RQ}\;\mathrm{weighted}\;\mathrm{equations}:\;RQ=\frac{VCO_{2}}{VO_{2}}$$

### 4.11. CODA Noninvasive Blood Pressure System

The blood pressure of the mice was measured by means of a modern high-precision and noninvasive method using the CODATM Standard system (Kent Scientific Corporation, Torrington, CT, USA). This method uses a specialized volume pressure recording (VPR) sensor that are placed over the animal’s tail to measure blood volume changes. The mice do need to be restrained and artificially heated in specific holders in order to maintain normal BP.

### 4.12. Statistical Analysis

Each experiment was repeated at least three times. The normal distribution of the data was assessed using the Kolmogorov–Smirnov test. The data are presented as the means ± SD with respect to the number of samples (*n*) in each group. Significance was assessed by comparing the mean values using Student’s *t*-test and the Wilcoxon rank sum test for independent groups as follows: * *p* < 0.05; ** *p* < 0.01; *** *p* < 0.001.

## 5. Conclusions

Together, our study provides novel insights into the gut microbiota-derived metabolites and underpins that gut flora-produced l-Histidine/ImP can protect against radiation-induced cardiopulmonary toxicity. Mechanistic investigation showed that the radioprotective function of l-Histidine or ImP is dependent of gut microbiota. Clinically, l-Histidine or ImP might be employed as a potential and promising radioprotective agent to fight against radiotherapy intertwined cardiopulmonary injury.

## Figures and Tables

**Figure 1 ijms-22-11436-f001:**
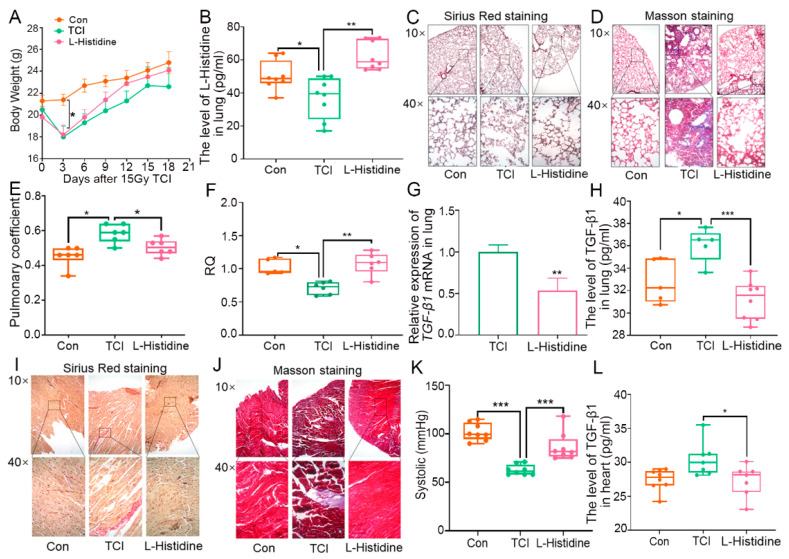
Gut microbiota-derived l-Histidine improves radiation toxicity in lungs and heart. (**A**) Body weight of mice after 15 Gy TCI and l-Histidine treated group after TCI. (**B**) The l-Histidine expression in lung tissues by ELISA (*n* = 8). (**C**,**D**) The lung tissues of each group mice were stained with Sirius Red (**C**) and Masson (**D**) and were observed under 10× and 40× microscopes, respectively. (**E**) Determination of the pulmonary coefficient in each group of mice (*n* = 6). (**F**) RQ of mice in each group of mice in 24 h (*n* = 6). (**G**,**H**) The TGF-β1 levels in lung tissues of each group mice were assessed by q-PCR and ELISA (*n* = 6). (**I**,**J**) The heart tissues of each group mice were stained with Sirius Red (**I**) and Masson (**J**) and were observed under 10× and 40× microscopes, respectively. (**K**) The systolic of each group of mice (*n* = 8). (**L**) The TGF-β1 levels in the heart tissues of each group mice were assessed by ELISA (*n* = 7). (The data are presented as mean ± SD, * *p* < 0.05, ** *p* < 0.01 and *** *p* < 0.001; Student’s *t*-test).

**Figure 2 ijms-22-11436-f002:**
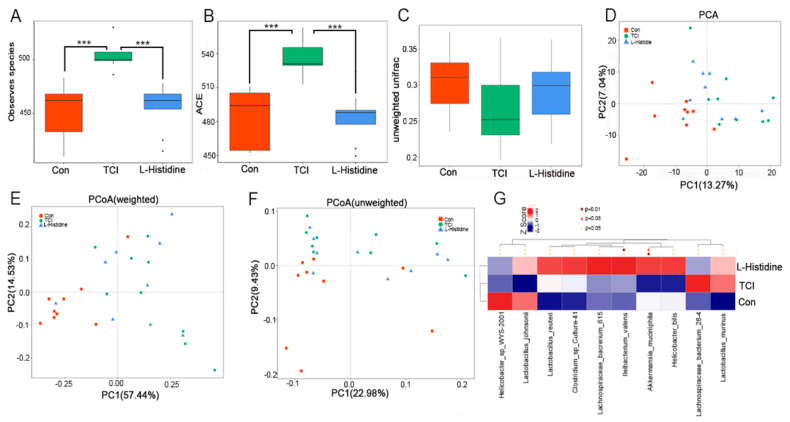
l-Histidine shapes the gut microbiota configuration after local chest irradiation. (**A**,**B**) The α-diversity of gut bacteria was measured by 16S rRNA high-throughput sequencing. In detail, the data were represented as the observed species number (**A**) and the ACE diversity index (**B**); (**C**,**D**) The β-diversity of enteric bacteria was compared by unweighted (**C**) and PCA (**D**) unifrac analysis. (**E**,**F**) PCoA were used to examine the alteration of the intestinal bacteria taxonomic pattern; (**G**) the heat map is color-based based on row Z-scores. The mice with the highest and lowest bacterial level are in red and blue, respectively. (*n* = 8 per group, the data are presented as mean ± SD, *** *p* < 0.001, Statistically significant differences are indicated: Wilcoxon rank sum test).

**Figure 3 ijms-22-11436-f003:**
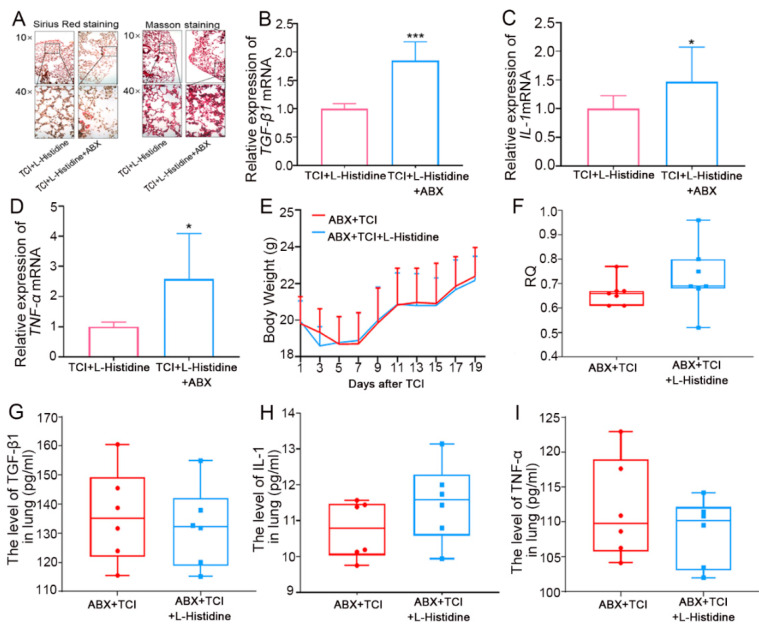
Gut microbiota contributes to l-Histidine-mediated radioprotection. (**A**) The lung tissues of each group mice were stained with Sirius Red and Masson in the ABX and l-Histidine and l-Histidine alone treated groups after TCI. (**B**–**D**) The TGF-β1, IL-1 and TNF-α levels in lung tissues of each group of mice were assessed by q-PCR. (**E**) The body weight of each group of mice (not statistically significant). (**F**) The RQ in each group mice after 24 h (*n* = 7). (**G**–**I**) The TGF-β1, IL-1 and TNF-α levels in lung tissues of each group mice are assessed by ELISA (*n* = 6). (The data were presented as mean ± SD, * *p* < 0.05 and *** *p* < 0.001; Student’s *t*-test).

**Figure 4 ijms-22-11436-f004:**
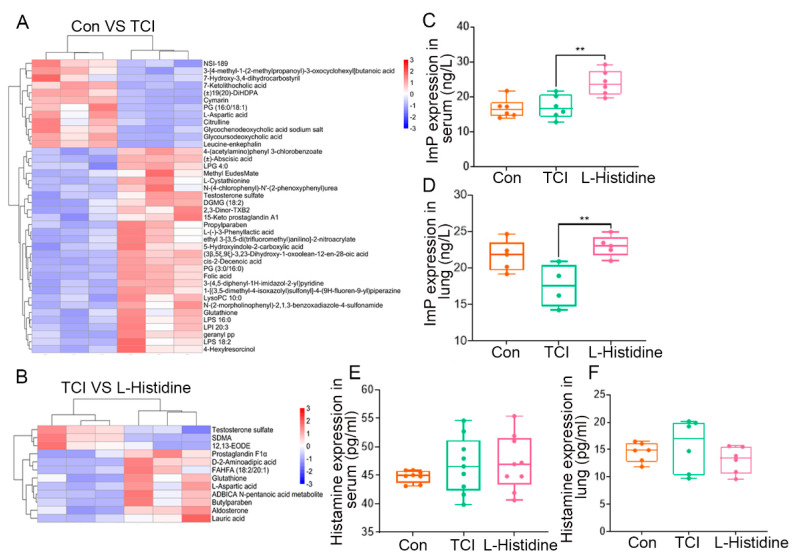
l-Histidine remolds the gut microbiota metabolome fluctuated by local chest irradiation. (**A**,**B**) Heatmap of the different identified metabolites in the fecal pellets from mice. In the volcano plots, each point represents a metabolite; (**A**) changes in the metabolites in the fecal pellets from mice in the TCI group and the control group; (**B**) changes in the metabolites in the fecal pellets from mice in the TCI group and l-Histidine group. (**C**,**D**) The ImP expression in serum (*n* = 6) and lung (*n* = 5) in each group. (**E**,**F**) The histamine expression in serum (*n* = 8) and lung (*n* = 6) in each group. (The data are presented as mean ± SD, ** *p* < 0.01; Student’s *t*-test).

**Figure 5 ijms-22-11436-f005:**
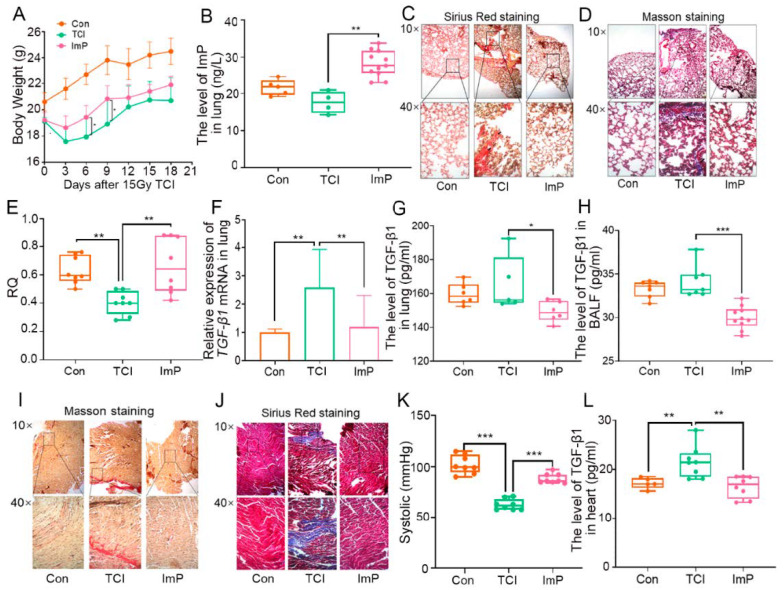
Imidazole propionate ameliorates chest local irradiation-induced toxicity. (**A**) Body weight of mice after 15 Gy. (**B**) The ImP expression in lung tissues by ELISA (*n* ≥ 6). (**C**,**D**) The lung tissues of each group mice were stained with Sirius Red (**C**) and Masson (**D**) and were observed under 10× and 40× microscopes, respectively. (**E**) Respiratory quotient (RQ) of mice in each group mice in 24 h (*n* = 8). (**F**,**G**) The TGF-β1 levels in lung tissues of each group mice were assessed by q-PCR and ELISA (*n* = 6). (**H**) The TGF-β1 levels in the BALF of each group of mice were assessed by ELISA (*n* = 8). (**I**,**J**) The heart tissues of each group of mice were stained with Sirius Red (**I**) and Masson (**J**) and were observed under 10× and 40× microscopes, respectively. (**K**) The systolic of each group of mice (*n* = 8). (**L**) The TGF-β1 levels in heart tissues of each group mice were assessed by ELISA (*n* = 8). (The data are presented as mean ± SD, * *p* < 0.05, ** *p* < 0.01 and *** *p* < 0.001; Student’s *t*-test).

**Figure 6 ijms-22-11436-f006:**
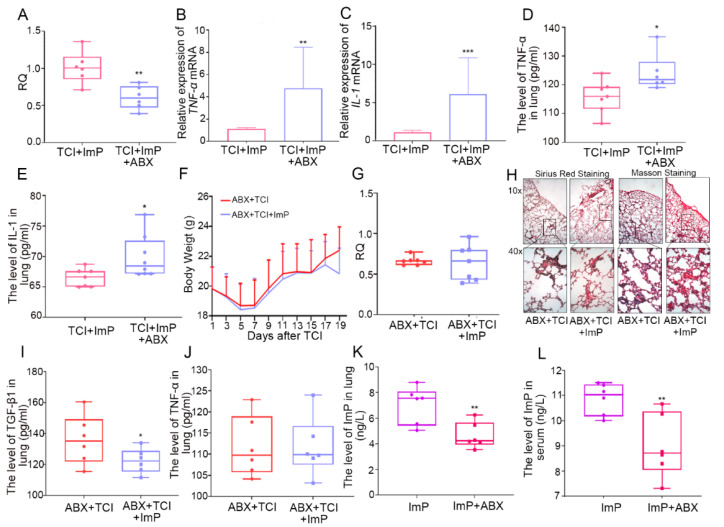
Gut microbiota impacts the assimilation of imidazole propionate. (**A**) The RQ in each group mice in 24 h (*n* = 6). (**B**,**C**) The TNF-α and IL-1 levels in the lung tissues of each group mice were assessed by q-PCR. (**D**,**E**) The TNF-α and IL-1 levels in lung tissues of each group mice were assessed by ELISA (*n* = 6). (**F**) The body weight of each group mice (not statistically significant). (**G**) The RQ in each group of mice after 24 h (*n* = 7). (**H**) The lung tissues of each group mice were stained with Sirius Red and Masson in the ABX-treated group after TCI. (**I**,**J**) The TGF-β1 and TNF-α levels in lung tissues of each group of mice were assessed by ELISA (*n* = 6). (**K**) The level of ImP in lung were assessed by ELISA (*n* = 6). (**L**) The level of ImP in serum were assessed by ELISA (*n* = 6). (The data are presented as mean ± SD, * *p* < 0.05, ** *p* < 0.01 and *** *p* < 0.001; Student’s *t*-test).

**Figure 7 ijms-22-11436-f007:**
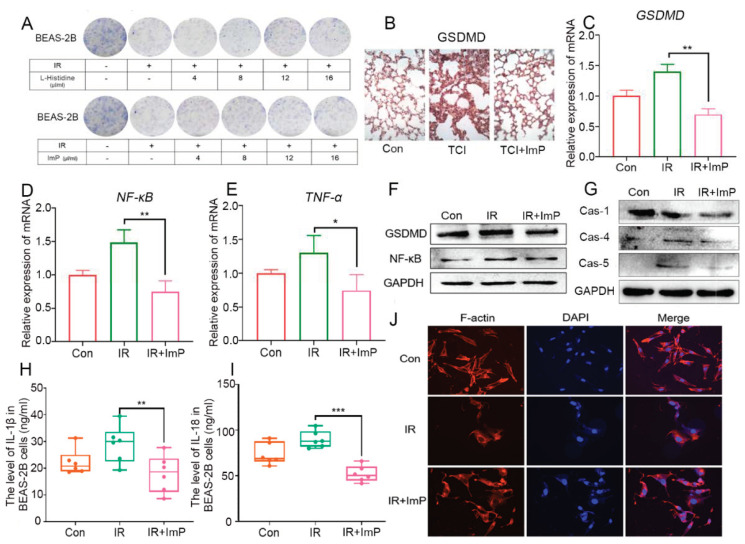
Imidazole propionate inhibits pyrolysis of irradiated lung cells. (**A**) The radiation sensitivity of BEAS-2B cells for l-Histidine and ImP in different densities were assessed through a clone formation experiment after radiation. (**B**) The GSDMD of irradiated lung tissues was analyzed by IHC. (**C**) The GSDMD of irradiated BEAS-2B cells was analyzed by qRT-PCR. (**D**,**E**) The relative expression of NF-κB and TNF-α mRNA in irradiated BEAS-2B cells were assessed by qRT-PCR (*n* = 6). (**F**) The GSDMD and NF-κB expression in irradiated BEAS-2B cells were examined by Western blotting. (**G**) The caspase-1, caspase-4, and caspase-5 expression in irradiated BEAS-2B cells were examined by Western blotting. (**H**,**I**) The level of IL-1β and IL-18 in irradiated BEAS-2B cells (*n* = 6). (**J**) Immunofluorescence showed the F-actin expression in irradiated BEAS-2B cells. (The data are presented as mean ± SD, * *p* < 0.05, ** *p* < 0.01 and *** *p* < 0.001; Student’s *t*-test).

## Data Availability

Not applicable.

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
