# Peer review of "Gut Microbiota-Derived l-Histidine/Imidazole Propionate Axis Fights against the Radiation-Induced Cardiopulmonary Injury"

_ijms, 2021, doi:10.3390/ijms222111436_

Round 1

Reviewer 1 Report

The present manuscript data indicate gut microbiota-derived L- Histidine/ImP protect against the cardiopulmonary complication caused by radiotherapy.  The content presented in this manuscript is exciting and sound, and will be interesting to the readers of this journal. Nevertheless, suggestions and comments are listed below to improve the manuscript.

-- Please highlight the revised areas for this reviewer to review them efficiently.

Overall:

  1. title page, title: insert “the” before radiation…
  2. page 3 Figure 1A. No statistical analysis with body weights is presented.
  3. page 5 Figure 3E. No statistical analysis with body weights is presented.
  4. The words of “radioprotective effects” and “radiomitigative effects” are freely used throughout the manuscript and create confusion to the reviewer whether the gut microbiota-derived L-Histidine and ImP were studied for pretreatment (i.e., Radioprotective effects) or posttreatment (i.e., Radiomitigative effects). Therefore, please clarify and use the right expression/phrases to describe your studies and results.
  5. page 8 Figure 6F. No statistical analysis with body weights is presented.
  6. page 10, line 136: Insert “the” before radiation-induced…
  7. page 10, line 137: Insert “the” before radiation-induced…
  8. page 10, line 146: Insert “the” before radiation-induced…
  9. page 11, line 179: Insert “the” before radiation-induced…
  10. page 11, line 209: Replace “cells” with “cell”.
  11. page 11, lines 211-213: Did you give the fecal dissolving fluid to the TCI group Before TCI or after TCI? How long ago did you give it if prior to TCI? Or how long after did you give it if post TCI? Please include the detailed information in the manuscript.
  12. page 12, lines 215-218: With antibiotics, how did you give them to mice after TCI? When did you start giving these antibiotics to mice after TCI? Please include the detailed information in the manuscript.
  13. Please include mean±sem or mean±SD for the data in each figure legend and which type of statistics was used in the Figure.

-- Again, please highlight the revised areas for this reviewer to spot them efficiently.

Author Response

Reply to comments

Dear Editor:

Thank you very much for your kind work and giving us an opportunity to revise our manuscript (IJMS-1401947) to International Journal of Molecular Sciences. Overall, the comments have been fair, encouraging and constructive. We have learned much from them. The comments are very helpful for revising and improving our manuscript. According to the reviewers’ comments, we carefully revised the manuscript. Please find the corrections and responds to reviewers’ comments point by point as follows:

Reviewer #1:

Question 1: title page, title: insert “the” before radiation…

Answer: Thank you very much for the suggestion. We inserted “the” before radiation in title. Writing modifications were provided in our revised manuscript.

Question 2: page 3 Figure 1A. No statistical analysis with body weights is presented.

Answer: Thank you very much for the suggestion. The statistical analysis was added to the figure legend of Figure 1A. Writing modifications were provided in our revised manuscript.

Question 3: page 5 Figure 3E. No statistical analysis with body weights is presented.

Answer: Thank you very much for the suggestion. The body weights of the experimental mice in Figure 3E represented no significant different. The statistical analysis was added to the figure legend. Writing modifications were provided in our revised manuscript.

Question 4: The words of radioprotective effects and radiomitigative effects are freely used throughout the manuscript and create confusion to the reviewer whether the gut microbiota-derived L-Histidine and ImP were studied for pretreatment (i.e., Radioprotective effects) or posttreatment (i.e., Radiomitigative effects). Therefore, please clarify and use the right expression/phrases to describe your studies and results.

Answer: Thank you very much for the suggestion. The comment is very helpful for our future study. In this study, the mice were pretreated with L-Histidine or ImP for 1 day before irradiation. Then, we assessed the effects of the gut microbiota-derived metabolites on radiation injuries. Thus, the phrase of “radioprotective effects” was more accurate for our study. We corrected the errors throughout the whole manuscript. Writing modifications were provided in our revised manuscript.

Question 5: page 8 Figure 6F. No statistical analysis with body weights is presented.

Answer: Thank you very much for the suggestion. Same to Figure 3E, the body weights of the experimental mice in Figure 6F represented no significant different. The statistical analysis was added to the figure legend. Writing modifications were provided in our revised manuscript.

Question 6: page 10, line 136: Insert the before radiation-induced

Answer: Thank you very much for the suggestion. We inserted “the” before radiation-induced in line 386. Writing modifications were provided in our revised manuscript.

Question 7: page 10, line 137: Insert the before radiation-induced

Answer: Thank you very much for the suggestion. We inserted “the” before radiation-induced in line 387. Writing modifications were provided in our revised manuscript.

Question 8: page 10, line 146: Insert the before radiation-induced

Answer: Thank you very much for the suggestion. We inserted “the” before radiation-induced in line 403. Writing modifications were provided in our revised manuscript.

Question 9: page 11, line 179: Insert the before radiation-induced

Answer: Thank you very much for the suggestion. We inserted “the” before radiation-induced in line 440. Writing modifications were provided in our revised manuscript.

Question 10: page 11, line 209: Replace cells with cell.

Answer: Thank you very much for the suggestion. We replaced “cells” with “cell” in line 98. Writing modifications were provided in our revised manuscript.

Question 11: page 11, lines 211-213: Did you give the fecal dissolving fluid to the TCI group Before TCI or after TCI? How long ago did you give it if prior to TCI? Or how long after did you give it if post TCI? Please include the detailed information in the manuscript.

Answer: Thank you very much for the suggestion. We provided a schematic diagram for FMT administration in supplementary information (Figure S5), and added more details about FMT in the method section. Figure and writing modifications were provided in our revised manuscript.

Question 12: page 12, lines 215-218: With antibiotics, how did you give them to mice after TCI? When did you start giving these antibiotics to mice after TCI? Please include the detailed information in the manuscript.

Answer: Thank you very much for the suggestion. Same to Question 11, we provided a schematic diagram for antibiotics administration in supplementary information (Figure S6), and added more details about antibiotics cocktail treatment in the method section. Figure and writing modifications were provided in our revised manuscript.

Question 13: Please include mean±sem or mean±SD for the data in each figure legend and which type of statistics was used in the Figure.

Answer: Thank you very much for the suggestion. We added mean±SD and the type of statistics in each figure legend. Writing modifications were provided in our revised manuscript.

Reviewer 2 Report

The manuscript (ijms-1401947) entitled “Gut microbiota-derived L-Histidine/imidazole propionate axis fights against radiation-induced cardiopulmonary injury” presents the protective effect of L-Histidine and Imidazole propionate against injuries induced by ionizing radiation in an attempt to demonstrate the important value that intestinal flora could have in the bioavailability of these substances and their possible clinical application. It is a current topic, novel and of great interest in the field of radiobiology and radiological protection.

 Major Considerations

- Authors must properly reorder the sections of the submitted manuscript. Section 4 Material and methods should be the second section after the Introduction.

- The authors do not mention in their manuscript important information that is necessary to understand the work they have done. It would basically consist in answering the question: Do the authors intend to describe the potential radioprotective effect of the tested substances L-Histidine and Imidazole propionate? Or Do the authors intend to show that these substances should be incorporated, produced or increased their bioavailability depending on the bacterial flora? In my opinion, both options are interesting.

L-Histidine is an amino acid that has been considered radioprotective in numerous studies. Imidazole is less studied but appears to inhibit ionizing radiation-induced DNA damage repair. Histamine behaves as a sensitizer in numerous studies. Because the discussion in the submitted manuscript is short, the authors should collect this background in their discussion as it would help to clarify the results of the studies.

- If the authors want to show the importance of flora in the bioavailability of these substances, they should clarify it in each of the Material and Methods sections, as well as highlight these aspects in the results presented.

The most specific part of the Discussion in this regard is found in lines 147-161. However, those described in lines 156-161 are very difficult to find in Results as it has been presented. It would be important for the authors to be able to highlight those aspects that denote the importance of action of the flora in the process of availability of the tested substances.

Minor considerations:

 There are numerous typographical errors in the submitted manuscript:

- Text numbering is lost on line 99 and reappears on line 100, but several pages later.

- There are numerous typographical errors lines 213 (10-d), 231 (H2O2) and 250 and following (CO2…).

- Place spaces between text and reference.

- References should also be checked

Author Response

Reply to comments

Dear Editor:

Thank you very much for your kind work and giving us an opportunity to revise our manuscript (IJMS-1401947) to International Journal of Molecular Sciences. Overall, the comments have been fair, encouraging and constructive. We have learned much from them. The comments are very helpful for revising and improving our manuscript. According to the reviewers’ comments, we carefully revised the manuscript. Please find the corrections and responds to reviewers’ comments point by point as follows:

Reviewer #2:

 Major Considerations

- Authors must properly reorder the sections of the submitted manuscript.

Question 1: Section 4 Material and methods should be the second section after the Introduction.

Answer: Thank you very much for the suggestion. We modified the section Material and methods to the second section in the revised manuscript.

Question 2: The authors do not mention in their manuscript important information that is necessary to understand the work they have done. It would basically consist in answering the question: Do the authors intend to describe the potential radioprotective effect of the tested substances L-Histidine and Imidazole propionate? Or Do the authors intend to show that these substances should be incorporated, produced or increased their bioavailability depending on the bacterial flora? In my opinion, both options are interesting.

Answer: Thank you very much for the suggestion. The comment is very helpful. In this study, we aimed screen effective radioprotective metabolites through the gut microbiome and provide new therapeutic option for the radiation-induced cardiopulmonary injury. We clarified our purpose in the introduction section (lines 69-70). In mechanistic investigation, we found that the radioprotective effects of L-Histidine and imidazole propionate were dependent on the intestinal microbiota. Thus, we added the relative evidence in the discussion section (lines 416-419). Writing modifications were provided in our revised manuscript.

Question 3: L-Histidine is an amino acid that has been considered radioprotective in numerous studies. Imidazole is less studied but appears to inhibit ionizing radiation-induced DNA damage repair. Histamine behaves as a sensitizer in numerous studies. Because the discussion in the submitted manuscript is short, the authors should collect this background in their discussion as it would help to clarify the results of the studies.

Answer: Thank you very much for the suggestion. The comment is very important. We collected the background of L-Histidine (lines 395-401) and imidazole propionate (lines 442-445) added the evidence in the discussion section. Writing modifications were provided in our revised manuscript.

Question 4: If the authors want to show the importance of flora in the bioavailability of these substances, they should clarify it in each of the Material and Methods sections, as well as highlight these aspects in the results presented.

Answer: Thank you very much for the suggestion. The gut microbiota was important for the bioavailability of L-Histidine and ImP, we clarify these substances in Material and methods section (lines 124-127) and Results section (parts 3.3 and 3.6). Writing modifications were provided in our revised manuscript.

Question 5. The most specific part of the Discussion in this regard is found in lines 147-161. However, those described in lines 156-161 are very difficult to find in Results as it has been presented. It would be important for the authors to be able to highlight those aspects that denote the importance of action of the flora in the process of availability of the tested substances.

Answer: Thank you very much for the suggestion. The comment is very helpful. We rewrote the relative sentences in Discussion section. Writing modifications were provided in our revised manuscript.

Minor considerations:

There are numerous typographical errors in the submitted manuscript:

Question 6.Text numbering is lost on line 99 and reappears on line 100, but several pages later.

Answer: Thank you very much for the suggestion. We corrected the error in the revised manuscript.

Question 7. There are numerous typographical errors lines 213 (10-d), 231 (H2O2) and 250 and following (CO2).

Answer: Thank you very much for the suggestion. We corrected all the typographical errors throughout the manuscript. Writing modifications were provided in our revised manuscript.

Question 8. Place spaces between text and reference.

Answer: Thank you very much for the suggestion. We added spaces between text and reference in the revised manuscript.

Question 9. References should also be checked.

Answer: Thank you very much for the suggestion. The references was checked. Writing modifications were provided in our revised manuscript.

Round 2

Reviewer 2 Report

The authors have responded to all the questions presented and have incorporated into the text the most important aspects that had been indicated.